# Nestin-Expressing Cells in the Lung: The Bad and the Good Parts

**DOI:** 10.3390/cells10123413

**Published:** 2021-12-04

**Authors:** Gilberto Jaramillo-Rangel, María-de-Lourdes Chávez-Briones, Adriana Ancer-Arellano, Marta Ortega-Martínez

**Affiliations:** Department of Pathology, School of Medicine, Autonomous University of Nuevo León, Monterrey 64460, Mexico; gilberto.jaramillorn@uanl.edu.mx (G.J.-R.); MDLOURDES.CHAVEZBRN@uanl.edu.mx (M.-d.-L.C.-B.); adar7035@gmail.com (A.A.-A.)

**Keywords:** nestin, lung, cancer, cell therapy, cancer stem cells, mesenchymal stem cells

## Abstract

Nestin is a member of the intermediate filament family, which is expressed in a variety of stem or progenitor cells as well as in several types of malignancies. Nestin might be involved in tissue homeostasis or repair, but its expression has also been associated with processes that lead to a poor prognosis in various types of cancer. In this article, we review the literature related to the effect of nestin expression in the lung. According to most of the reports in the literature, nestin expression in lung cancer leads to an aggressive phenotype and resistance to chemotherapy as well as radiation treatments due to the upregulation of phenomena such as cell proliferation, angiogenesis, and metastasis. Furthermore, nestin may be involved in the pathogenesis of some non-cancer-related lung diseases. On the other hand, evidence also indicates that nestin-positive cells may have a role in lung homeostasis and be capable of generating various types of lung tissues. More research is necessary to establish the true value of nestin expression as a prognostic factor and therapeutic target in lung cancer in addition to its usefulness in therapeutic approaches for pulmonary diseases.

## 1. Introduction

Cytoskeletal proteins include microtubules, actin filaments, and intermediate filaments (IFs). IF proteins share a similar structure, consistent with a central α-helical domain flanked by head (N-terminus) and tail (C-terminus) domains. According to their amino acid sequence and gene structure, IFs were initially grouped into five subtypes. Group VI was created after the discovery of nestin (an acronym for neuroepithelial stem cell protein). Several characteristics were decisive for nestin to be classified as the only member of a different group: a truncated head domain, its unique α-helical region, an extremely long C-terminal end, and the presence of a third intron in its gene (Figure 1). Owing to these structural features, especially its small head domain, nestin cannot form homo-oligomers, but it can form heteropolymeric filaments with other IF proteins, such as vimentin and glial fibrillary acidic protein (GFAP) [1,2].

Nestin was originally described in neuroepithelial stem cells of developing and adult brains and, since then, its expression has been described in many types of stem/progenitor cells in tissues such as the bone marrow, heart, skeletal muscle, pancreas, testis, and hair. In these and other sites, nestin might have a role in active proliferation, wound healing, and tissue regeneration. On the other hand, several studies have reported that nestin is a marker of cancer stem cells (CSCs, Figure 2) expressed in malignancies of organs such as the brain, uterus, cervix, prostate, ovary, testis, and pancreas. In the presentation and progression of cancer, nestin expression has been associated with several processes, including cell proliferation, angiogenesis, and metastasis [3].

Thus, nestin might have contrasting effects on cell function and fate. Until now, it has not been fully understood which cellular factors (e.g., cell type, developmental stage) determine the effect of nestin expression. However, nestin, as a member of the cytoskeleton, contributes to the regulation of the spatial localization of cell organelles, in addition to its participation in cell signaling pathways that regulate phenomena such as differentiation, proliferation, apoptosis, motility, oxidative stress, and migration [4,5].

Despite the importance of the lung (for example, lung cancer is the leading cause of cancer-related deaths worldwide) and nestin in current medicine, there is relatively little information about their interaction. In this article, we review the literature related to the effect of nestin expression in the lung.

## 2. Nestin in the Prenatal Lung

Thyroid transcription factor-1 (TTF-1) is expressed in the lung from the beginning of organogenesis until adulthood. TTF-1 plays a fundamental role in the lung-specific expression of surfactant proteins and the Clara cell secretory protein. In 2001, Lonigro et al. [6] showed that TTF-1 might play a role in the organogenesis of the forebrain through the upregulation of nestin. However, they did not investigate whether there was any interaction between TTF-1 and nestin in the prenatal lung.

In fact, it was not until four years later that the first reports of the presence of nestin expression in the prenatal lung began. Rieske et al. [7] demonstrated that human fetal lung-derived fibroblasts (MRC-5 cells) have characteristics of embryonic stem cells and cells of neuroectodermal origin, including nestin expression. Additionally, Yu et al. [8], using transgenic mice expressing 3β-hydroxysteroid Δ7-reductase (DHCR7) driven by a murine nestin promoter, demonstrated the expression of DHCR7 only in the brains and lungs of neonatal transgenic animals. Interestingly, when Dubois et al. [9] created the Nes-Cre1 transgenic mouse line, in which Cre recombinase expression is driven by the rat nestin promoter and the rat nestin intron 2 enhancer, the authors found that while by embryonic day 15.5 recombination occurred in practically all cells of the nervous system, little or no recombination occurred in organs such as the heart, thymus, liver, and lung. However, in these organs, recombination was observed around blood vessels, perhaps due to recombination in endothelial cells, which are known to express nestin [10].

Hua et al. [11] isolated human fetal lung cells, which they characterized as mesenchymal stem cells (MSCs), and induced them to differentiate into lineages of adipocytes, osteoblasts, and neural cells; even into sperm-like cells. The neural cells were positive for myelin basic protein, GFAP, and nestin. Similarly, Hu et al. [12] isolated cells from fetal bovine lung tissues that were morphologically consistent with fibroblasts, expressed surface markers of MSCs, and were induced to undergo osteogenic and neurogenic differentiation. Neuronal cells were positive for microtubule-associated protein 2 and nestin expression.

Qiao and Chu [13] detected positive nestin expression in the kidneys, intestines, stomachs, colons, spinal cords, and lungs of embryos and fetuses, both normal and with placenta previa. The expression of nestin was higher in placenta previa than in normal fetuses, and the authors concluded that nestin expression might be correlated with the clinicopathological characteristics of placenta previa. Nestin expression decreased throughout the prenatal development process. Specifically, in the lung, nestin-positive cells were observed mainly around the breathing lumen, microvascular walls, and interstitial areas. No nestin expression was found in the luminal epithelial cells of blood vessels.

Finally, although it is not entirely clear whether nestin functions in lung organogenesis, experimental evidence from murine models indicates that, as a member of the IF family, nestin may play a role in alveolar septation [14].

## 3. Nestin-Expressing Cells in the Adult Lung

### 3.1. Potential Use of Nestin-Expressing Cells in Cell Therapy

Sabatini et al. [15] isolated human adult bronchial fibroblast-like cells, which express most of the surface antigen characteristics of MSCs. These cells also express nestin and are able to differentiate along adipogenic, chondrogenic, and osteogenic lineages when cultured under appropriate inducible conditions. In addition, they produce the extracellular matrix proteins collagen and fibronectin. Ricciardi et al. [16] obtained similar results regarding nestin expression and differentiation potential when they isolated MSCs from lung explants obtained from living organ donors. Furthermore, these authors found that MSCs reside at the perivascular level and are capable of supporting B cell growth in addition to downregulating T and NK cell proliferation. MSCs have been shown to be effective in the treatment of many diseases, including cardiac ischemia, diabetes, neurological disorders, and bone as well as cartilage diseases [17,18].

According to findings described in several papers, nestin-positive cells might have important roles in lung homeostasis. We analyzed the presence of apoptosis and the expression of proliferating-cell nuclear antigen (PCNA) in addition to nestin in lung cartilage sections of adult CD1 mice. Apoptosis and PCNA were detected in several chondrocytes. Serial section analysis demonstrated that PCNA-positive cells differed from cells in apoptosis, indicating that cell turnover was occurring. Nestin-positive cells were observed in connective tissue associated with cartilage as well as in perivascular cells [19]. Then, we found nestin-positive cells inside of cartilage islets, and cells in division were very close to these cells [20]. These findings indicate that nestin-positive cells in adults are able to differentiate into lung chondrocytes, despite the widespread belief that healthy adult chondrocytes normally maintain a stable resting phenotype and resist differentiation and proliferation throughout life [21]. Additionally, these findings improve current knowledge about cartilage biology and could provide new cell candidates for cartilage tissue engineering. In another study [22] we detected nestin-positive cells in perivascular areas and in connective tissues that were in close proximity to the lung airway epithelium. These cells were also detected among the cells lining the epithelium. It is likely that nestin-positive stem cells localize in the lung airway epithelium to participate in its normal turnover. Since we previously reported similar findings in lung cartilage, pluripotent nestin-positive cells capable of generating various types of lung tissue could exist, which may provide novel approaches to therapy for devastating pulmonary diseases.

Zhao and Burt [23] carried out immunohistochemical staining of normal tissues from the heart, pancreas, kidney, liver, colon, and lung obtained from rats, mice, guinea pigs, and pigs. Nestin was one of the antigens investigated. Interestingly, these authors did not report nestin expression in the lung. There is no clear explanation for the variability in the detection of nestin observed among studies. This could be due to methodological issues or other factors that require further investigation. It is also possible that the cells that express nestin are so scarce that they can sometimes go unnoticed. The low availability of stem cells could be a limitation for their use in cell therapy. Sha et al. [24] stimulated A549 lung cancer cells to display stem-cell-like characteristics (including nestin expression) using a combination of five small molecules. This strategy could provide a potential therapeutic approach for use in the treatment of lung diseases, including cancer.

### 3.2. Nestin-Expressing Cells in Lung Diseases

Birbrair et al. [25] reported the existence of two pericyte subtypes: type I, which is negative for nestin expression, and type II, which expresses nestin. They found that type I pericytes proliferate and accumulate at injured sites in the brain, spinal cord, heart, kidney, and lung. After injury, type I pericytes differ from scar-forming PDGFR β+ cells in the brain and spinal cord; they produce collagen in the lung but not in the kidney or heart. These findings indicate that pericyte subtypes respond differentially to tissue injury and that the production of collagen by type I pericytes is organ-dependent. Kishaba et al. [26] found nestin-positive fibroblast-like cells in the stroma of the kidney, skin, pancreas, and lung under fibrosing conditions. Complete characterization of the mechanisms underlying scar formation may reveal ways to improve tissue repair under pathological conditions.

Pulmonary hypertension (PH) includes a heterogeneous group of conditions that involve the proliferation and remodeling of the pulmonary arterial circulation, resulting in an increase in mean pulmonary pressure. The most common cause of PH is left-sided heart disease [27,28]. Chabot et al. [29] induced PH in rats through exposure to hypobaric hypoxia. PH was characterized by alveolar thickening, reactive fibrosis, and vascular structural abnormalities. An increase in nestin mRNA and protein levels was observed. A transition of endothelial and epithelial cells into a mesenchymal phenotype associated with nestin expression was observed. Using three different animal models and human samples of PH, Saboor et al. [30] found that the expression of nestin was significantly enhanced in the lungs early during the development of PH and localized to proliferating vascular smooth muscle cells (VSMCs). Similar findings were reported by Veteskova et al. [31]: mRNA levels of nestin increased in VSMCs when they induced PH in rats by administering monocrotaline (MCT). Bhagwani et al. [32] found endothelial cells positive for nestin expression in the remodeled pulmonary arteries of PH patients and of a rat model of PH. They then showed that the transient overexpression of nestin promotes endothelial proliferation and angiogenesis in vitro. Finally, Zhou et al. [33] showed increased nestin expression in pulmonary artery smooth muscle cells in congenital heart-disease-associated PH patients and in an MCT plus aortocaval shunt-induced rat model of PH. This overexpression contributed to aberrant cell proliferation and pulmonary vascular remodeling. Taken together, data from these studies indicate that nestin might represent a novel diagnostic tool and potential therapeutic target for PH.

Chabot et al. [34] tested the hypothesis that nestin is expressed in lung fibroblasts and that its pattern of expression represents a distinct marker of pulmonary remodeling secondary to myocardial infarction (MI) and type I diabetes. They found that in the lungs of post-MI rats, a reactive fibrotic response was associated with a concomitant increase in nestin protein/mRNA levels, despite the absence of PH. Therefore, based on its established proliferative function, nestin upregulation in lung fibroblasts may have directly contributed to the initiation of the reactive fibrotic response in post-MI rat lungs. By contrast, in the lungs of streptozotocin-induced type I diabetic rats, the absence of a reactive fibrotic response was associated with the significant downregulation of nestin mRNA/protein expression. This study showed that a disparity in nestin regulation characterized the early pattern of lung remodeling secondary to MI and type I diabetes. Based on antecedents, such as those described in previous work, Hertig et al. [35] conducted a study in which they demonstrated nestin upregulation in the hypertrophied/fibrotic left ventricle of adult rats with a constricted suprarenal abdominal aorta. This overexpression was attributed to the increased number of interstitial collagen type I cells coexpressing nestin. Furthermore, a subpopulation of nestin (+) cells was detected in the adventitial region of predominantly large-caliber blood vessels, although their identity and role in the reactive fibrotic response remain unknown.

On the other hand, nestin could also participate in the pathogenesis of lung diseases in which there is no direct relationship with the heart, such as those described in the preceding paragraphs. Thies et al. [36] found that any expression of nestin in chemo-naïve biopsies and chemo-treated surgical specimens from patients with malignant pleural mesothelioma (MPM) was associated with decreased survival. Nestin allows further prognostic stratification among histologic variants of MPM.

Using nestin-Cre; ROSA26-EYFP mice, in which nestin-positive cells and their progeny express enhanced yellow fluorescent protein (EYFP), Ke et al. [37] found in a chronic asthma model that Ras homolog family member A/Rho-associated protein kinase 1 (RhoA/ROCK) signaling drives MSC differentiation toward fibroblasts/myofibroblasts and enhances airway remodeling, whereas the inactivation of this signaling pathway promotes MSC differentiation into epithelial cells for airway repair. These findings deepened our understanding of the pathological mechanisms of asthma and provided a new therapeutic target for patients with this disease.

Accidental or intentional ingestion of paraquat (PQ) can cause fatal poisoning. The ingestion of PQ causes acute respiratory distress syndrome due to uncontrolled massive matrix production in the lung, resulting in progressive fibrosis. Using an animal model, Zhang et al. [38] showed that one of the mechanisms by which PQ causes lung damage is the rapid activation of nestin and MSCs, which contribute to the production of the extracellular matrix.

Finally, nestin is known to play an important role in various aspects related to lung cancer. However, this information deserves a separate section.

## 4. Nestin-Expressing Cells in Lung Cancer

In 2020, lung cancer was the second most commonly diagnosed cancer (surpassed only by female breast cancer), with an estimated 2.2 million new cases, and the leading cause of cancer-related death, with an estimated 1.8 million deaths [39]. However, although nestin is known to have a role in various malignancies, there is relatively little information in the literature about the effect of nestin expression in lung cancer.

Chen et al. [40] investigated nestin expression in cancer and adjacent normal tissues from 52 patients with non-small-cell lung cancer (NSCLC) by immunohistochemical staining. Of the 52 patient samples, 26 were adenocarcinomas (ADCs), 25 were squamous-cell carcinomas (SCCs), and 1 was large-cell carcinoma (LCC). Nestin was observed in most tumor cells and was significantly associated with poor differentiation, ADC, lymph node metastasis, microvessel density, and lymphatic vessel density.

Janikova et al. [41] analyzed nestin expression in the archived paraffin blocks of 121 NSCLC patient samples by immunohistochemical staining. Of the 121 patient samples, 82 were SCCs and 39 were ADCs. Nestin was detected in the epithelium in 66% and the vasculature in 70% of patients. No relationship between nestin expression and disease-free survival (DFS) and/or overall survival (OS) was found.

In another study [42], nestin was analyzed immunohistochemically in 171 consecutive patients with NSCLC (131 with ADCs, 31 with SCCs, and 9 with others), and the effect of nestin expression on clinicopathological parameters and survival was evaluated. Nestin positivity was observed in 15.8% of patients; positivity was significantly associated with SCCs, poor differentiation, intratumoral vascular invasion, lymph node metastasis, intratumoral lymphatic invasion, pleural invasion, and a poor prognosis. Additionally, nestin expression increased the risk of death after adjusting for other clinicopathological parameters. Nestin positivity was observed not only in tumor cells but also in vascular endothelial cells and in tumor stromal fibroblasts.

Skarda et al. [43] examined the immunohistochemical expression of nestin in 114 NSCLC patients (78 with SCCs and 37 with ADCs). Nestin was detected in the epithelium in 35.1% of patients and in the vasculature in 7.9% of patients. Nestin positivity did not significantly correspond to DFS or OS.

Nestin expression was examined immunohistochemically in 71 NSCLC patients (35 with ADCs, 34 with SCCs, and 2 with LCCs). Nestin-positive cells were found in the majority of samples, and a significant association of nestin expression with the subset of NSCLC patients displaying poor outcomes and high levels of proliferative markers (PCNA and Ki-67) was observed. On the other hand, the knockdown of nestin expression in NSCLC cell lines A549, H1299, and H460 significantly inhibited cell proliferation, decreased colony forming ability, and arrested the G1/S cell cycle [44].

Liu et al. [45] recruited 153 patients with NSCLC (66 with ADCs, 64 with SCCs, and 23 with others). Nestin expression in tumor samples was determined by immunohistochemical staining. Nestin positivity was related to tumor differentiation and lymphatic metastasis, and was an independent prognostic factor for OS after controlling for confounding factors. Moreover, the knockout of nestin in A549 and H1299 cell lines enhanced cancer cell apoptosis and inhibited cell proliferation and invasion and colony formation. Finally, nestin depletion increased E-cadherin expression while inhibiting vimentin, suggesting that nestin modulates epithelial–mesenchymal transition (EMT) to promote cell invasion, resulting in tumor metastasis.

Nestin was analyzed immunohistochemically in 90 NSCLC patients (66 with ADCs, 18 with SCCs, and 6 with others) receiving adjuvant platinum-based chemotherapy. Nestin positivity was observed in 31.1% of patients and was associated with the loss of E-cadherin expression/vimentin-positive expression (EMT) as well as a poor prognosis. Multivariable analysis confirmed that nestin expression was an independent prognostic indicator in those patients [46].

In another study [47], immunohistochemical analysis of surgical specimens revealed nestin expression in 20 of 48 (41.7%) ADC cases and in 25 of 47 (53.2%) SCC cases. Nestin immunoreactivity significantly correlated with tumor size, lymph node metastasis, and poor survival in patients with ADCs. High and moderate expression levels of nestin were confirmed in several lung ADC cell lines, such as H1975 and PC-3. Nestin knockout decreased proliferation, migration, invasion, and sphere formation in ADC cells, whereas nestin upregulation via nestin gene transfection resulted in the opposite changes.

Ahmed et al. [48] evaluated nestin expression by real-time PCR, serum vascular endothelial growth factor (VEGF), and Bcl-2 by ELISA in 27 ADC patients and 15 control subjects. Nestin was significantly higher in ADC patients than in control subjects and significantly correlated with VEGF and Bcl-2. This may be attributed to the proangiogenic and antiapoptotic roles of nestin, because VEGF is an important angiogenic factor and Bcl-2 has an antiapoptotic function.

Nestin expression was examined immunohistochemically in 371 surgically resected NSCLC specimens. Nestin positivity was associated with poor tumor differentiation and/or an increased proliferation index, and was an unfavorable prognostic marker for ADCs [49].

The expression and function of nestin have also been analyzed in other subtypes of lung cancer, such as small cell lung cancer (SCLC). The expression of nestin was studied in 21 lung cancer cell lines by Western blot and was detected immunohistochemically in surgically resected SCLC primary tumors (two samples) and metastatic SCLC tumors obtained from autopsies (two samples). To assess the function of nestin, its expression was silenced in two SCLC cell lines. Nestin was expressed in nine out of ten SCLC cell lines. Its expression level was significantly higher in SCLC cell lines than in NSCLC cell lines. Nestin-knockdown cells exhibited decreased cell invasion and proliferation capabilities. Finally, nestin was detected in SCLC tumor cells and tumor vessels in all clinical specimens [50].

Using nestin-knockdown cells and nestin-overexpressing cells, Sone et al. [51] examined the relationship between nestin expression and cell proliferation, as well as chemosensitivity, in vitro and in vivo. Additionally, they analyzed nestin expression in drug-resistant lung cancer cell lines. Finally, they immunohistochemically examined nestin staining in samples from 84 SCLC patients. Nestin expression correlated positively with cell proliferation and negatively with chemosensitivity, and was upregulated in drug-resistant cell lines compared to their parental cells. Twenty-four patients (28.6%) harbored nestin-positive tumors. There were no significant differences in the response rate or progression-free survival (PFS) between nestin-positive and nestin-negative patients following first-line chemotherapy. In terms of second-line chemotherapy, patients with nestin-positive expression experienced a significantly shorter PFS.

Nestin expression was immunohistochemically studied in 30 patients with resected large-cell neuroendocrine carcinoma (LCNEC), and its associations with clinicopathological parameters, the Ki-67 labeling index (LI), and TTF-1 expression were evaluated. Nestin positivity was observed in eight of the thirty (26.7%) samples (in tumor cells, vascular endothelial cells, and fibroblasts in the tumor stroma) and was significantly associated with a high Ki-67 LI. Additionally, nestin expression was significantly associated with a poor prognosis [52].

In another study [53], immunohistochemical staining for nestin was performed in 88 patients with neuroendocrine lung tumors, 48 of which were LCNECs, 32 were typical carcinoids (TCs), and 8 were atypical carcinoids (ACs). Nestin expression was significantly higher in LCNEC than in TC and AC. Lymph node metastases, nestin expression, and patient age were independent negative prognostic factors.

Interestingly, when Bromińska et al. [54] analyzed immunohistochemical staining for nestin in 35 LCNEC, 15 TC, and 5 AC biopsies, they found that nestin expression did not correlate with lymph node metastasis, maximum size of the lesion, Ki-67 expression, or survival. The authors attributed these results to the relatively small and heterogeneous study group.

Since neuroendocrine tumors of the lung account for only 5% of all cases of lung cancer, only a few cell lines of these tumors exist. Windmöller et al. [55] reported the in vitro propagation and characterization of CSCs from a case of TC. These cells were positive for the expression of the CSC markers CD44, CD133, and nestin.

Several meta-analyses have examined the effect of nestin expression on lung cancer. A meta-analysis by Zhong et al. [56] revealed that reports from the literature have shown a significant association between nestin expression and stage in various types of cancer (pancreatic, prostatic, pulmonary, gastric, and oral cancers), especially lung cancer.

Li et al. [57] performed a meta-analysis to investigate the association of nestin expression with clinicopathological features and OS in NSCLC patients. They found that nestin expression was significantly associated with lymph node metastasis, stage, and OS in NSCLC patients.

Another meta-analysis was performed to determine the association between nestin and OS in NSCLC patients. The results indicated that the expression of nestin was significantly associated with OS in NSCLC patients and that the association was maintained in Asians and Caucasians after stratification by race. Additionally, the association was maintained when the meta-analysis was limited to studies that controlled for clinical parameters [58].

Finally, in a meta-analysis performed by Li et al. [59], nestin expression was significantly associated with unfavorable outcomes of differentiation degree, lymphatic metastasis, stage, tumor size, and poor OS. Associations were maintained in the subgroups stratified by patient origin, statistical analysis, follow-up periods, and in ADC as well as LCNEC cases, but not in SCC cases.

Table 1 summarizes the main clinical characteristics of lung cancer associated and not associated with the expression of nestin.

### How Does Nestin Work in Lung Cancer?

Most of the papers reviewed in the previous section presented evidence of an important role of nestin in the presentation and progression of lung cancer. Observations in several malignancies suggest that nestin may serve as a central organizer of important processes for cancer behavior. Nestin expression in tumor tissue has been associated with the degree of tumor differentiation and proliferation, the invasive and metastatic capacities of tumors and/or the degree of neoangiogenesis [60,61,62].

Several studies have reported that nestin is a marker of CSCs in different malignancies [63,64,65]. CSCs represent a distinct population of undifferentiated cells responsible for tumor initiation and maintenance, and play a prominent role in the growth, migration, and invasion of neoplasms. CSCs have the capacities of tumorigenicity, unlimited proliferation, self-renewal, multipotency, quiescence, and DNA repair, all of which can affect the effectiveness of chemotherapy and radiation on cancer treatments [66,67,68].

In addition to the characteristics of the CSCs described above, nestin-expressing cells have other properties associated with their roles in cancer. Studies involving IF functions in human diseases and normal tissue homeostasis have revealed that IF proteins serve multiple nonmechanical functions, including the regulation of apoptosis [1,4,69]. Resistance to cell death is an important characteristic of tumor cells to proliferate. Nestin has been found to have an antiapoptotic function through the regulation of various cell signaling pathways [70,71,72]. This antiapoptotic role of nestin has been reported in lung-cancer-related studies [45,48]. However, since the inhibition of apoptosis by nestin has not been observed in all types of cancer [73], more research is needed.

EMT is a process during which epithelial cells acquire stem cell characteristics. The hallmarks of EMT include the acquisition of a spindle-like/fibroblast morphology, the downregulation of epithelial cell surface markers and cytoskeleton components, the upregulation of mesenchymal markers and extracellular matrix components, and the upregulation and/or nuclear translocation of specific transcription factors [74,75]. EMT is involved in fundamental cellular processes, including the proliferation, metastasis, and drug resistance of tumors. Furthermore, EMT may be involved in the development of CSCs [76,77]. Several studies have presented evidence that nestin is involved in the molecular processes that lead to the presentation of EMT in different types of cancer, including lung cancer [45,46,78].

Tumor cells may migrate away from a primary site to inhabit new microenvironments, disrupting normal organ function in a phenomenon called metastasis [79]. Tumor angiogenesis is an important factor for the metastasis of neoplasms; the degree of metastasis and angiogenesis is associated with tumor aggressiveness and clinical outcomes [80]. Nestin expression has been reported to correlate with metastasis and angiogenesis in various tumors [81].

The metastasis of tumors from several organs to the lung has been analyzed by detecting nestin expression. Ackermann et al. [82] reported the generation of a transgenic mouse line that developed melanotic melanomas with high penetrance and a metastatic phenotype. Primary melanomas were transplanted subcutaneously into nude mice and, when injected intravenously into NOD/SCID mice, colonized the lung. Nestin-expressing cells were detected in primary melanomas and metastases as well as in experimental lung metastases of NOD/SCID mice. In another study, human neuroblastoma cell lines were injected subcutaneously into SCID mice, and their growth behavior and nestin as well as CD44 expression were analyzed. Most neuroblastoma cell lines produce metastatic deposits in the lung. Though nestin expression was not associated with the growth and metastatic behavior of the cells, CD44 expression reflected the specific metastatic properties of neuroblastoma cells [83]. With regard to breast cancer, while Meisen et al. [84] analyzed a cohort of 166 patients and found that nestin expression was significantly associated with the metastasis of breast cancer to the lung and brain, Sihto et al. [85] analyzed 234 patients and found that breast cancers that metastasize to the lungs infrequently express nestin. Finally, the ability of other cancers, such as pancreatic ductal adenocarcinoma [86] and prostate cancer [60], to metastasize to the lung has been positively associated with nestin expression. On the other hand, several of the articles reviewed in the previous section presented evidence of a role of nestin in the ability of lung cancer to metastasize to other organs [40,42,45,47,50,53].

Tumor angiogenesis is necessary for the establishment of metastases in distant organs. Nestin has been proposed as a reliable marker for proliferative endothelial cells in tissues undergoing neovascularization [87,88,89,90]. In fact, nestin expression has been detected in tumor angiogenesis in several cancer types, including astrocytoma [91], ependymoma [92], glioblastoma [93], melanoma [94], pancreatic cancer [95], gastric cancer [96], colorectal cancer [97], hepatocellular carcinoma [98], breast cancer [99], ovarian cancer [100], prostate cancer [101], and lung cancer [40,41,42,43,48,50,52,102]. Thus, it is clear that the expression of nestin is important for the process of angiogenesis. However, the mechanisms underlying the expression of nestin in angiogenesis are unclear [103]. Research related to this field has focused on experimental therapies to inhibit or prevent nestin-mediated angiogenesis [104,105,106]. The existence of animal models of nestin-dependent vascular formation [107,108] could be of great use to obtain new knowledge that fills the existing gaps.

The molecular details involved in the role of nestin in tumor behavior still require further clarification. Nestin participates in cell signaling pathways that regulate important cell phenomena. CDK5 is a kinase involved in tumorigenesis, with functions ranging from cell proliferation and apoptosis to invasion and angiogenesis [109]. Nestin protects CSCs from apoptosis by binding to CDK5, thereby inhibiting its pro-apoptotic function [3]. Additionally, nestin regulates the Akt/mTOR pathway, which has a critical role in autophagy, one of the mechanisms a cell utilizes to resist cell death [44]. Furthermore, nestin regulates the Wnt/β-catenin signaling pathway, which is critical to promote the proliferation and invasiveness of CSCs [1]. On the other hand, the nestin gene has specific regions in the second intron to which N-myc transcription factor binds. N-myc enhances nestin expression and cell proliferation as well as motility [110]. The combined activation of the nestin gene by Notch and KRAS causes the proliferation of nestin/PCNA-positive cells in some lesions, which may represent premalignant stages of tumorigenesis [111]. p53 is another transcription factor that interacts with the nestin gene. Nestin expression is kept in check by p53 in normal cells, and the formation of certain cancer types through the depletion of p53 levels was dependent on nestin expression [112]. Finally, although nestin expression in endothelial cells has been implicated in angiogenesis, the underlying mechanisms are unclear (see previous paragraph).

## 5. Conclusions

Nestin is a class VI IF protein expressed during development in a small subset of cells and tissues in normal adults under various pathological conditions. The expression of nestin, both in normal and abnormal tissue, indicates primitiveness, an un-differentiated state, plasticity, and a great proliferative potential. The aim of the present article was to review the literature related to the effect of nestin expression in the lung. The evidence collected so far indicates that nestin may be involved in very important aspects of lung cancer, including cell proliferation and differentiation, the inhibition of apoptosis, EMT, angiogenesis, and metastasis, which may lead to a low effectiveness of chemotherapy and radiation treatments with a consequent poor prognosis. Thus, nestin may serve as a prognostic factor and therapeutic target in lung cancer. On the other hand, evidence also indicates that nestin-positive cells may have a role in tissue homeostasis and be capable of generating various types of lung tissues, which may provide novel therapeutic approaches for devastating pulmonary diseases. Little is known about the underlying mechanisms and the regulatory signaling pathways that control nestin expression and function. More research is needed to understand the contrasting effects of nestin expression in the lung to obtain the most benefit from this information.

## Figures and Tables

**Figure 1 cells-10-03413-f001:**
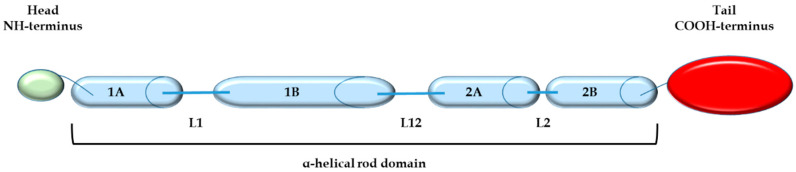
Nestin’s structure consists of a central α-helical rod domain (blue) flanked by globular NH (green) and COOH (red) domains. The α-helical rod domain consists of segments 1A and 1B separated by linker L1 and segments 2A and 2B with linker L2 between them, separated from 1B by linker L12.

**Figure 2 cells-10-03413-f002:**
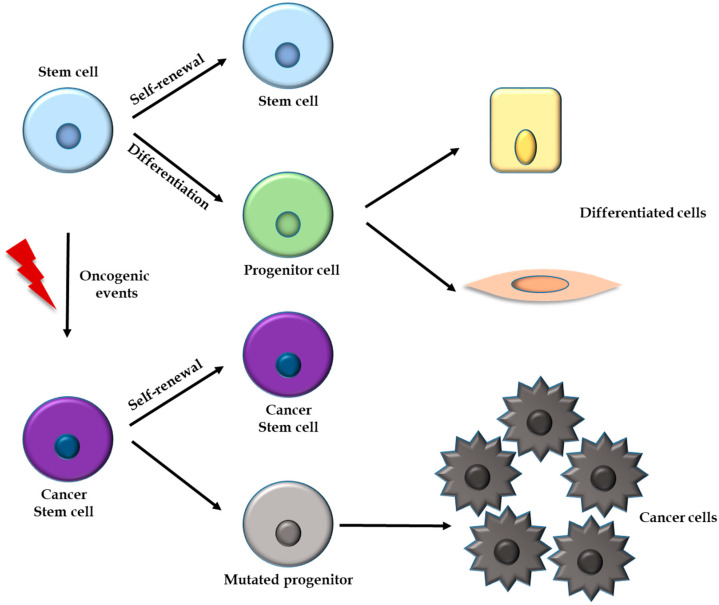
Stem cells are capable of generating more cells of the same type by self-renewal and progenitor cells by differentiation, which, in turn, originate several types of differentiated cells. Cancer stem cells (CSCs) are generated from normal stem cells after epigenetic mutations occur. Oncogenic events include the activation of proto-oncogenes in addition to the inactivation of tumor suppressor genes and DNA repair mechanisms. CSCs are capable of self-renewal and generating daughter cells (mutated progenitors) that have more limited self-renewal ability and can originate cancer cells.

**Table 1 cells-10-03413-t001:** Main clinical characteristics of lung cancer associated and not associated with the expression of nestin.

Reference	Lung Cancer Histology	Nestin Associated with	Nestin Not Associated with
[40]	NSCLC -ADC -SCC -LCC	Poor differentiation Lymph node metastasis High microvessel density High lymphatic vessel density ADC	
[41]	NSCLC -ADC -SCC		Disease-free survival Overall survival
[42]	NSCLC -ADC -SCC -Others	Poor differentiation Lymph node metastasis Intratumoral vascular invasion Intratumoral lymphatic invasion Pleural invasion Poor prognosis SCC	
[43]	NSCLC -ADC -SCC		Disease-free survival Overall survival
[44]	NSCLC -ADC -SCC -LCC	Poor outcome High levels of proliferative markers (PCNA and Ki-67)	
[45]	NSCLC -ADC -SCC -Others	Low tumor differentiation Lymph node metastasis	
[46]	NSCLC -ADC -SCC -Others	Epithelial-mesenchymal transition Poor prognosis	
[47]	NSCLC -ADC -SCC	In ADC: High tumor size Lymph node metastasis Poor survival	
[48]	NSCLC -ADC	High VEGF High Bcl-2	
[49]	NSCLC -ADC -SCC -LCC -Others	Poor differentiation Increased proliferation Poor prognosis in ADC	
[51]	SCLC	Response to second-line chemotherapy	Response to first-line chemotherapy
[52]	LCNEC	Poor prognosis High Ki-67 labeling index	
[53]	LCNEC TC AC	LCNEC Low overall survival	Lymph node metastasis
[54]	LCNEC TC AC		Lymph node metastasis Tumor size Ki-67 expression Overall survival
[56] *	Several subtypes reported by other authors	Median/advanced cancer stage	
[57] *	NSCLC	Median/advanced cancer stage Lymph node metastasis Low overall survival	
[58] *	NSCLC	Low overall survival	
[59] *	NSCLC -ADC -SCC LCNEC	Low tumor differentiation Median/advanced cancer stage Lymph node metastasis Low overall survival High tumor size	

NSCLC: non-small-cell lung cancer; ADC: adenocarcinoma; SCC: squamous-cell carcinoma; LCC: large-cell carcinoma; PCNA: proliferating-cell nuclear antigen; VEGF: vascular endothelial growth factor; SCLC: small-cell lung cancer; LCNEC: large-cell neuroendocrine carcinoma; TCs: typical carcinoids; ACs: atypical carcinoids; and * meta-analysis.

## Data Availability

Data sharing is not applicable to this review paper.

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
