# Peer review of "Nestin-Expressing Cells in the Lung: The Bad and the Good Parts"

_cells, 2021, doi:10.3390/cells10123413_

Round 1
Reviewer 1 Report
Authors have submitted “Nestin-expressing cells in the lung: the bad and the good parts” they aim to outline the data on the Nestin expressing cells in lung as prognostic marker in lung carcinoma. However, some major issues should be clarified:
Figure 2 should be better described more details e.g., oncogenic events in stem cells should be given.
In the introduction, the consensual brief definitions of somatic stem cells (SSCs), cancer stem cells (CSC), progenitor cells, side population (SP), main population (MP) cells, clonogenic cells, colony-forming unit (CFU), label-retaining cells (LRC), to help the readers understand their meaning within the biologic/hierarchical “frame” of normal and cancer stem cells. There is no information how many features nestin-positive cells are common with adult mesenchymal stem cells. Moreover, nestin is not typical biomarker of mesenchymal stem cells (MSCs), so nestin can not be used for MSCs identification. The role of nestin–positive cells in lung cancer is unclear, because in chapter 4 (Nestin-expressing cells in lung cancer) the authors only discuss immunohistochemical data concerning nestin expression in relation to clinico-pathological parameters of lung cancer.
The chapter 4.1 (How does nestin work in lung cancer?) the authors present only basic knowledge on carcinoma cells stem.
Conclusion need to be increased with more related text insights on the matter.
Author Response
Point 1: Authors have submitted “Nestin-expressing cells in the lung: the bad and the good parts” they aim to outline the data on the Nestin expressing cells in lung as prognostic marker in lung carcinoma. However, some major issues should be clarified:
Figure 2 should be better described more details e.g., oncogenic events in stem cells should be given.
In the introduction, the consensual brief definitions of somatic stem cells (SSCs), cancer stem cells (CSC), progenitor cells, side population (SP), main population (MP) cells, clonogenic cells, colony-forming unit (CFU), label-retaining cells (LRC), to help the readers understand their meaning within the biologic/hierarchical “frame” of normal and cancer stem cells.

Response 1: With the exception of cancer stem cells and progenitor cells, these terms were not used in the article, as they are not relevant or necessary for the purpose of the review. In this new version of the manuscript, we include more details in the description of Figure 2 (including what is related to oncogenic events), and at the same time we highlight the differences between cancer stem cells and progenitor cells. Please see the legend for Figure 2.
Point 2: There is no information how many features nestin-positive cells are common with adult mesenchymal stem cells. Moreover, nestin is not typical biomarker of mesenchymal stem cells (MSCs), so nestin cannot be used for MSCs identification.
Response 2: Although it is true that nestin is not a “typical” marker for mesenchymal stem cells (MSCs), many authors report that MSCs express nestin. Specifically for the lung, please see lines 97-100, 119-125, 130-141, 219-224, and 227-232 (note that line numbers apply when tracked changes are hidden in the text).
Point 3: The role of nestin–positive cells in lung cancer is unclear, because in chapter 4 (Nestin-expressing cells in lung cancer) the authors only discuss immunohistochemical data concerning nestin expression in relation to clinico-pathological parameters of lung cancer.
Response 3: The aim of the article was to review the literature related to the effect of nestin expression in the lung. What we describe in the article is what can be found in the literature. On the other hand, some authors describe the use of other approaches to analyze the relationship between nestin and lung cancer. Please see lines 269-271, 276-280, 291-293, 304-312, and 313-319 (note that line numbers apply when tracked changes are hidden in the text).
Point 4: The chapter 4.1 (How does nestin work in lung cancer?) the authors present only basic knowledge on carcinoma cells stem.
Response 4: In this new version of the manuscript we add more information to chapter 4.1. Please see lines 450-466 (note that line numbers apply when tracked changes are hidden in the text).
Point 5: Conclusion need to be increased with more related text insights on the matter.
Response 5: In this new version of the manuscript we add more information to the Conclusions. Please see lines 470-474 and 482-484 (note that line numbers apply when tracked changes are hidden in the text).
Note: in this new version of the manuscript we added references 109-112.
Reviewer 2 Report
The manuscript has been carefully reviewed. The current manuscript comprehensively reviewed literatures which characterized nestin-expressing cells either in the normal lung or in the diseased status. While agreeing the manuscript is potentially informative to the related field. However, the work lacks mechanismic discussion which can further support the important role of nestin expression in lung development or in lung diseases. Authors may consider improving the manuscript based on addressing whether expression of the intermediated filiments-nestin indeed contribute to lung disease progression or whether nestin expression is a consequence of the progression of lung disease.
Author Response
The manuscript has been carefully reviewed. The current manuscript comprehensively reviewed literatures which characterized nestin-expressing cells either in the normal lung or in the diseased status. While agreeing the manuscript is potentially informative to the related field. However, the work lacks mechanismic discussion which can further support the important role of nestin expression in lung development or in lung diseases. Authors may consider improving the manuscript based on addressing whether expression of the intermediated filiments-nestin indeed contribute to lung disease progression or whether nestin expression is a consequence of the progression of lung disease.

Response 1: The requested information was added to chapter 4.1. Please see lines 450-466 (note that line numbers apply when tracked changes are hidden in the text).
Note: in this new version of the manuscript we added references 109-112.
Reviewer 3 Report
In the present review, Jaramillo-Rangel et al. provide an overview on the role of Nestin-expressing cells in the prenatal lung, adult lung, lung diseases, and lung cancer. In particular, the authors discuss how nestin may work in cancer.
The paper is well-written, and the authors highlight the proposed issues in a comprehensive way. Nevertheless, the following minor issue has to be addressed before considering it suitable for publication.
Minor issue
Figure 2 does not match well with the text. Authors state “On the other hand, several studies have reported that nestin is a marker of cancer stem cells (CSCs) expressed in malignancies of organs such as the brain, uterus, cervix, prostate, ovary, testis, and pancreas (Figure 2).” Figure 2 shows what the CSCs are. I suggest editing the text or the figure.
Author Response
In the present review, Jaramillo-Rangel et al. provide an overview on the role of Nestin-expressing cells in the prenatal lung, adult lung, lung diseases, and lung cancer. In particular, the authors discuss how nestin may work in cancer.
The paper is well-written, and the authors highlight the proposed issues in a comprehensive way. Nevertheless, the following minor issue has to be addressed before considering it suitable for publication.
Minor issue
Figure 2 does not match well with the text. Authors state “On the other hand, several studies have reported that nestin is a marker of cancer stem cells (CSCs) expressed in malignancies of organs such as the brain, uterus, cervix, prostate, ovary, testis, and pancreas (Figure 2).” Figure 2 shows what the CSCs are. I suggest editing the text or the figure.
Response 1: We edit both the text and the figure. Please see (with change tracking feature turned on) the second paragraph of the Introduction and the legend for Figure 2.
Round 2
Reviewer 1 Report
No comments and remarks to the present version of manuscript: “Nestin-expressing cells in the lung: the bad and the good parts”.